# A Phase Ib Clinical Trial of Metformin and Chloroquine in Patients with *IDH1*-Mutated Solid Tumors

**DOI:** 10.3390/cancers13102474

**Published:** 2021-05-19

**Authors:** Mohammed Khurshed, Remco J. Molenaar, Myra E. van Linde, Ron A. Mathôt, Eduard A. Struys, Tom van Wezel, Cornelis J. F. van Noorden, Heinz-Josef Klümpen, Judith V. M. G. Bovée, Johanna W. Wilmink

**Affiliations:** 1Department of Medical Oncology, Cancer Center Amsterdam, Amsterdam UMC location AMC, University of Amsterdam, 1105 AZ Amsterdam, The Netherlands; m.khurshed@amsterdamumc.nl (M.K.); r.j.molenaar@amsterdamumc.nl (R.J.M.); m.vanlinde@amsterdamumc.nl (M.E.v.L.); h.klumpen@amsterdamumc.nl (H.-J.K.); 2Department of Medical Biology, Cancer Center Amsterdam, Amsterdam UMC location AMC, University of Amsterdam, 1105 AZ Amsterdam, The Netherlands; c.j.vannoorden@amsterdamumc.nl; 3Department of Clinical Pharmacology, Cancer Center Amsterdam, Amsterdam UMC location AMC, University of Amsterdam, 1105 AZ Amsterdam, The Netherlands; r.mathot@amsterdamumc.nl; 4Department of Clinical Chemistry, Cancer Center Amsterdam, Amsterdam UMC location VU, University Medical Center, 1081 HV Amsterdam, The Netherlands; e.struys@amsterdamumc.nl; 5Department of Pathology, Leiden University Medical Center, 2311 EZ Leiden, The Netherlands; T.van_Wezel@lumc.nl (T.v.W.); J.V.M.G.Bovee@lumc.nl (J.V.M.G.B.); 6Department of Genetic Toxicology and Cancer Biology, National Institute of Biology, 1000 Ljubljana, Slovenia

**Keywords:** metformin, chloroquine, cancer, isocitrate dehydrogenase, pharmacokinetics, glioblastoma, intrahepatic cholangiocarcinoma, chondrosarcoma

## Abstract

**Simple Summary:**

Mutations in the isocitrate dehydrogenase 1 (*IDH1*) gene occur in high-grade chondrosarcoma, high-grade glioma and intrahepatic cholangiocarcinoma. Due to the lack of effective treatment options, these aggressive types of cancer have a dismal outcome. The metabolism of *IDH1*-mutated cancer cells is reprogrammed in order to produce the oncometabolite *D*-2-hydroxyglutarate (*D*-2HG). In this clinical trial, we used the oral antidiabetic drug metformin and the oral antimalarial drug chloroquine to disrupt the vulnerable metabolism of *IDH1*-mutated solid tumors. We found that the combination regimen of metformin and chloroquine is well tolerated, but the combination did not induce a clinical response in this patient population. Secondly, we confirmed the clinical usefulness of *D/L*-2HG ratios in serum as a biomarker and the ddPCR-facilitated detection of an IDH1 mutation in circulating DNA from peripheral blood.

**Abstract:**

Background: Mutations in isocitrate dehydrogenase 1 (*IDH1*) occur in 60% of chondrosarcoma, 80% of WHO grade II-IV glioma and 20% of intrahepatic cholangiocarcinoma. These solid *IDH1*-mutated tumors produce the oncometabolite *D*-2-hydroxyglutarate (*D*-2HG) and are more vulnerable to disruption of their metabolism. Methods: Patients with *IDH1*-mutated chondrosarcoma, glioma and intrahepatic cholangiocarcinoma received oral combinational treatment with the antidiabetic drug metformin and the antimalarial drug chloroquine. The primary objective was to determine the occurrence of dose-limiting toxicities (DLTs) and the maximum tolerated dose (MTD). Radiological and biochemical tumor responses to metformin and chloroquine were investigated using CT/MRI scans and magnetic resonance spectroscopy (MRS) measurements of *D*-2HG levels in serum. Results: Seventeen patients received study treatment for a median duration of 43 days (range: 7–74 days). Of twelve evaluable patients, 10 patients discontinued study medication because of progressive disease and two patients due to toxicity. None of the patients experienced a DLT. The MTD was determined to be 1500 mg of metformin two times a day and 200 mg of chloroquine once a day. A serum *D/L*-2HG ratio of ≥4.5 predicted the presence of an *IDH1* mutation with a sensitivity of 90% and a specificity of 100%. By utilization of digital droplet PCR on plasma samples, we were able to detect tumor-specific *IDH1* hotspot mutations in circulating tumor DNA (ctDNA) in investigated patients. Conclusion: Treatment of advanced *IDH1*-mutated solid tumors with metformin and chloroquine was well tolerated but did not induce a clinical response in this phase Ib clinical trial.

## 1. Introduction

Somatic heterozygous mutations in *IDH1* and *IDH2* occur in up to 60% of central conventional chondrosarcoma, 80% of WHO grade II–IV glioma, 20% of intrahepatic cholangiocarcinoma and 10% of acute myeloid leukemias (AML) [1]. In glioma, patients with an *IDH1* mutation have a relatively prolonged survival, whereas prognosis of other *IDH1/2*-mutated tumors remains poor. *IDH1* mutations are present in a large fraction or even all cancer cells in glioma [2], rendering *IDH1* mutations an interesting target for treatment, because the high molecular homogeneity diminishes the risk of therapy resistance [3]. It has been shown that *IDH1* mutations sensitize cancer cells in vitro to therapies that involve oxidative stress, such as radiotherapy, cisplatin and carmustine [4,5,6]. Appreciation of the causative role of *IDH1/2* mutations in oncogenesis led to the development of the *IDH1*-mutation inhibitor agent ivosidenib (AG-120) [7] and the *IDH2*-mutation inhibitor enasidenib (AG-221). Both are FDA-approved for the treatment of relapsed or refractory *IDH1*-mutated or *IDH2*-mutated AML [8,9].

IDH1 and IDH2 are homodimeric enzymes that reversibly convert isocitrate to α-ketoglutarate (αKG) in the cytoplasm and mitochondria, respectively. Hotspot mutations in *IDH1*, of which *IDH1*^R132H^ is the most prevalent in glioma and chondrosarcoma, lead to heterodimeric enzymes (*IDH1^WT/MUT^*), loss of wild-type IDH1 enzyme function and a neomorphic IDH1 activity that converts αKG into the oncometabolite *D*-2-hydroxyglutarate (*D*-2HG) [10]. *D*-2HG exerts its oncogenic effects via competitive inhibition of αKG-dependent dioxygenases [11,12], that are essential for epigenetic regulation of gene expression, including that of metabolic genes [13,14,15]. Moreover, *D*-2HG serum concentrations may serve as a surrogate biomarker of the presence of an *IDH1/2* mutation [8,16,17,18].

Reprogramming of cellular metabolism is one of the hallmarks of cancer [14,19,20]. *IDH1/2* mutations affect carbohydrate and NADP^+^/NADPH metabolism by causing loss of wild-type IDH1/2 function. Metabolic pathways are rewired and carbon metabolites are redirected away from the tricarboxylic acid (TCA) cycle for production of *D*-2HG [14,21,22,23,24]. *IDH1*-mutated cancer cells are more vulnerable to inhibition of metabolism with inhibitors of the electron transport chain (ETC), such as metformin that inhibits complex I of the ETC [4,25]. In vitro, we previously showed that *IDH1*-mutated cancer cells are more sensitive to metformin compared to *IDH1* wild-type cancer cells [4]. However, sensitivity to metformin in chondrosarcoma cell lines was independent of the presence of an *IDH1* mutation [26]. Moreover, metformin has attracted interest as an anti-cancer drug [27], since an association between metformin use in type 2 diabetes mellitus (T2DM) patients and a reduced risk of breast, colon, pancreas and prostate cancer was found [28,29,30,31,32,33], as well as a reduced risk of mortality, as compared with patients treated with insulin or sulfonylureas [34].

A second consequence of rewired metabolism in *IDH1*-mutated cells is the dependence on the glutaminolysis pathway, which provides anaplerosis to the TCA cycle at the level of αKG for generation of the oncometabolite *D*-2HG [6,14,21,35]. In this alternative pathway, the final step of glutaminolysis can be inhibited by chloroquine, an antimalarial drug, as well as metformin targeting glutamate dehydrogenase [20,36,37,38]. In addition, *IDH1*-mutated glioma cells in metabolic stress show increased levels of autophagy in order to provide substrates for energy production [39]. These anti-cancer properties of chloroquine in combination with inhibition of autophagy by chloroquine may thus be selective for *IDH1*-mutated cells because it inhibits glutaminolysis and autophagy on which these cells are dependent [40]. Chondrosarcoma cell lines are sensitive to chloroquine because it suppresses autophagy, irrespective of the *IDH1*-mutation status [26]. Both chloroquine and metformin are cheap, widely used and readily available drugs with a safety profile that is favorable comparable to most chemotherapeutic agents.

The present clinical trial investigated the safety and maximum tolerated dose (MTD) of the combination therapy of metformin and chloroquine in patients with *IDH1*-mutated chondrosarcoma, glioma and intrahepatic cholangiocarcinoma. As secondary objectives, the pharmacokinetics of metformin and chloroquine combinational treatment and efficacy were assessed by measuring tumor size and/or levels of *D*-2HG and *L*-2HG in serum to determine their ratio.

## 2. Materials and Methods

### 2.1. Patient Population

Eligible patients were aged ≥18 years with *IDH1/2*-mutated newly-diagnosed, recurrent, relapsed or refractory grade II–III chondrosarcoma, WHO grade II–IV glioma or intrahepatic cholangiocarcinoma. The tumor material (obtained by surgery performed before or after this clinical trial (ClinicalTrials.gov NCT02496741) in the context of regular patient care or a tumor biopsy specifically for this clinical trial) had to carry a neomorphic *D*-2HG-generating mutation in *IDH1* or *IDH2* as determined by sequencing of tumor DNA or as shown by elevated *D*-2HG-to-*L*-2HG ratios in mass spectrometry (MS) of serum or magnetic resonance spectroscopy (MRS) imaging of the tumor. Other eligibility criteria included a WHO performance status ≤2, adequate renal function (creatinine <150 μmol/L or a creatinine clearance >60 mL/L), liver function (bilirubin <1.5 times the normal upper limit; ALAT and ASAT <2.5 times the normal upper limit), and bone marrow function (white blood cells >3.0 × 10^9^/L, platelets >100 × 10^9^/L). Patients were not eligible when they were concomitantly treated with other anti-cancer therapies (i.e., chemotherapy, targeted therapy, radiation therapy). Palliative therapy was permitted, such as local palliative radiotherapy, dexamethasone, and non-enzyme-inducing anti-epileptic drugs (with the exception of topiramate) in patients with glioma and epileptic seizures. Furthermore, patients were not eligible in case they were known to have a serious concomitant medical condition or used interacting medication that could not be replaced, a known hypersensitivity to either metformin or chloroquine, or when they had been treated with either metformin or chloroquine for another indication in the previous 6 months.

### 2.2. Study Design

This was a phase Ib, open-label, dose-escalation study to assess the safety, dose-limiting toxicities (DLTs), maximum tolerated dose (MTD) and the pharmacokinetic interactions of the combination of metformin and chloroquine in patients with *IDH1/2*-mutated chondrosarcoma, glioma and intrahepatic cholangiocarcinoma. The study was conducted at the Amsterdam University Medical Centers in The Netherlands. The study was executed in a standard 3 + 3 dose-escalation fashion: at least three patients per dose level were recruited, and the dose level was to be expanded to six patients when 1/3 patients experienced a DLT. Dose escalation to the next dose level was permitted when a DLT occurred in 0/3 or in ≤1/6 patients. In case of DLT(s) in ≥1/3 or in ≥2/6 patients, that dose level was declared intolerable. A DLT was defined as any of the following events related to study treatment occurred during the first treatment cycle, as defined by National Cancer Institute Common Terminology Criteria for Adverse Events version 4.03 (CTCAE): neutropenia grade 4 or febrile neutropenia grade 3 (fever ≥38.5 °C), grade ≥4 trombocytopenia or grade 3 trombocytopenia with bleeding, rash grade ≥2, diarrhea grade ≥3 or any other treatment-related toxicity grade ≥3, and missing >7 days of treatment for toxicity reasons, all despite optimal supportive care.

### 2.3. Study Treatment

To enable pharmacokinetic analyses between metformin and chloroquine, patients were treated with single-agent metformin in the first week before chloroquine was added to their treatment. In week 1, 500 mg of metformin was given once a day during the first 5 days. Subsequently, the metformin dose was escalated as outlined in Table 1. This escalation schedule is based on an earlier phase II clinical trial in pancreatic cancer [27]. The purpose of the lower metformin starting dose was to reduce gastrointestinal side effects of metformin and mimics recommended dosage schedules of metformin treatment in patients with T2DM. Treatment with chloroquine started on day 8. The chloroquine dose was fixed and was not escalated during the study.

### 2.4. Recommended Phase II Dose

The MTD is the dose at which ≥2/3 or ≥2/6 patients experienced a DLT. One dose level below the MTD, or dose level 3 in case of 0/6 DLTs at that final dose level, was considered the recommended dose (RD) for follow-up phase II clinical trials. Three patients were observed for 4 weeks at a dose level before buildup to the next dose level started. When a patient was withdrawn from the study prior to completing 28 days of therapy without experiencing a DLT, an additional patient was added to that dose level. Patients missing 7 or more doses due to toxicity were not replaced since these patients were considered to have experienced a DLT.

### 2.5. Pharmacokinetics

Pharmacokinetics and pharmacokinetic interactions between metformin and chloroquine were monitored and assessed in order to evaluate a relationship between drug exposure, toxicity and/or efficacy. Predose plasma levels were determined in blood samples obtained prior to study medication ingestion on day 8 (week 2), day 29 (week 5) and at the end of the study. Since chloroquine administration started on day 8, plasma samples on that day contained a metformin plasma concentration that reflected metformin monotherapy. The pharmacokinetic interactions between metformin and chloroquine were evaluated by comparison of the metformin concentration on day 8 with the metformin concentration on subsequent time points. The relationship between exposure and toxicity was evaluated in all samples.

### 2.6. Detection of D-2HG Levels

Previous studies demonstrated that circulating total 2HG was a surrogate biomarker for an *IDH1/2*-mutation status. In order to distinguish *D*-2HG (which is more specific for *IDH1/2* mutations) from the less specific *L*-2HG, we detected *D*-2HG and *L*-2HG levels in patient serum using liquid chromatography-tandem mass spectrometry (LC-MS/MS). Our method uses a chiral derivatizing agent, (+)-di-O-acetyl-l-tartaric anhydride (DATAN), to modify the *D* and *L*-stereoisomers of 2HG, allowing separation and quantification by LC-MS/MS [41,42,43]. The *D*-2HG and *L*-2HG levels are shown as the 2HG enantiomeric ratio [41].

### 2.7. Therapy Response

Response was assessed using Response Evaluation Criteria In Solid Tumors (RECIST) 1.1 guidelines [44] for chondrosarcoma and cholangiocarcinoma or Response Assessment in Neuro-Oncology (RANO) guidelines [45] for glioma on images obtained with CT or MRI scans. Scans were performed at screening and every 8 weeks after study inclusion.

### 2.8. ctDNA Analysis with the Digital Droplet Polymerase Chain Reaction

To quantify the variants in the ctDNA isolated from plasma, digital droplet polymerase chain reaction (ddPCR) assays with 20× primers with FAM- and HEX-labelled hydrolysis probes were used according to the manufacturer’s protocol. The ddPCR supermix for probes (no dUTP) and the assay were mixed in 20 μL with 20 ng ctDNA in a semi-skirted ddPCR 96-well plate. The droplets were generated using the QX200 droplet generator. The 96-well plate containing the droplets was sealed with pierceable heat seal and placed in the T100 Thermal Cycler. The PCR program was started with initial denaturation for 10 min at 95 °C followed by 40 cycles: 10 s at 94 °C and 30 s at 55 °C. The PCR program was ended by cooling down to 4 °C overnight. Positive and negative droplets were measured using a QX200 Droplet Reader (all reagents and machines for these measurements were purchased from Bio-Rad Laboratories, Veenendaal, The Netherlands).

### 2.9. Statistical Analysis

The occurrence of adverse events and clinical outcomes were described non-quantitatively. *p* Values were calculated as described in the figure legends with a significance level cutoff of α = 0.05. Significance levels are shown by * (*p* < 0.05), ** (*p* < 0.01), *** (*p* < 0.001) and **** (*p* < 0.0001). Data were processed in Excel version 2016 for Windows (Microsoft, Redmond, WA, USA) and using GraphPad Prism version 8.0.0 for Windows (GraphPad Software, San Diego, CA, USA).

## 3. Results

### 3.1. Characterization of the Study Cohort

The flowchart in Figure 1 shows that 38 patients with eligible tumor histologies were pre-screened for the presence of an *IDH1/2* mutation between November 2015 and May 2019. In total, 32 cholangiocarcinoma, 3 glioma and 3 chondrosarcoma patients underwent the pre-screening. Twenty patients had an *IDH1* mutation, including 14 cholangiocarcinoma, 3 glioma and 3 chondrosarcoma patients. Of these patients, 3 patients did not meet the study inclusion criteria; one patient with cholangiocarcinoma because of ongoing use of metformin for the treatment of T2DM, one patient with glioma because of participation in another clinical trial, and one cholangiocarcinoma patient because of unsolved hyperbilirubinemia (Figure 1). The high rate of 53% of the patients with an *IDH1* mutation in the pre-screened group is an exaggeration of the true prevalence of the *IDH1* mutation, since patients with *IDH1/2* mutation were referred to enroll in the study. The most frequent *IDH1* mutations found were R132C (50%), R132H (21%) and R132G (14%). We identified no patients with an *IDH2* mutation. Table 2 summarizes the baseline characteristics of the 17 eligible patients that were enrolled in the study. All patients except one had received one or more lines of systemic treatment prior to study enrolment.

### 3.2. Safety and Dose Adjustments

Seventeen patients started the study treatment (Figure 1). Twelve patients received at least 4 weeks of study treatment and were thus evaluable for toxicity assessments (Table 3). Patients remained in the study for a median duration of 43 days (range: 7–74 days). Five patients discontinued study participation during the first 4 weeks of treatment, in 4 cases because of clinical progression and in one case due to toxicity. Of the 12 evaluable patients who received metformin and chloroquine, 10 patients discontinued because of progressive disease and 2 patients due to toxicity. None of the patients experienced a DLT. The treatment-related adverse events per dose level are listed in Table 4. All observed treatment-related adverse events were CTCAE-grade ≤2 toxicities. The most frequently reported clinical toxicities of any grade included nausea (28%), anorexia (23%), fatigue (16%), diarrhea (13%) and vomiting (10%). In two patients, the metformin dose was de-escalated from 2000 mg per day to 1000 mg per day; in one due to toxicity without meeting DLT criteria; and in one due to abdominal pain, which was related to progressive disease. One patient with back pain due to a bone metastasis underwent local palliative radiotherapy during study participation. Regarding serious adverse events (SAE), one patient with glioma had to be hospitalized due to hydrocephalus and one patient with cholangiocarcinoma had to be hospitalized due to cholangitis (treated with antibiotics, no intervention). Both SAEs were considered to be unrelated to the study medication. The MTD was determined as 1500 mg metformin two times a day and 200 mg chloroquine once a day. The study protocol specified that this highest dose level was the MTD as well as the RD. According to protocol, we expanded this dose level to six patients. Because all patients showed clinical or radiological progression after eight weeks of study treatment, we considered it unethical to enroll three additional patients at this dose level. We determined the RD for future clinical trials with this combination to be 1500 mg metformin two times a day and 200 mg chloroquine once a day.

### 3.3. Pharmacokinetics

Blood concentrations of metformin and chloroquine were obtained from all evaluable patients and are shown in Figure 2. We observed a dose-level dependent increase of the plasma metformin concentration; 0.86 ± 0.32 mg/L, 1.86 ± 0.24 mg/L and 2.38 ± 0.38 mg/L, after administration of dose levels 1, 2 and 3, respectively (*p* = 0.0015). As expected with non-escalating chloroquine doses, whole-blood chloroquine concentrations were stable at subsequent time points with 520.7 ± 306.3 μg/L at week 4 and 462.5 ± 194.5 μg/L at week 8. The plasma metformin concentration was comparable between single-agent administration and co-administration with chloroquine (Appendix A).

### 3.4. Plasma D-2HG Concentrations

Previous studies demonstrated that circulating total 2HG was a surrogate biomarker for an *IDH1/2* mutation status. In order to distinguish *D*-2HG (which is more specific for *IDH1/2* mutations) from the less specific *L*-2HG, we calculated the *D/L*-2HG ratio in serum to determine a cutoff marker value for the presence of an *IDH1* mutation [41]. Analyzing the metabolite concentration in a subset of the screened 38 patients, the median value of total 2HG and *D*-2HG serum concentration was 5.6 ± 1.3 µmol/L and 5.3 ± 1.2 µmol/L in patients with an *IDH1*-mutated tumor and 1.0 ± 0.2 µmol/L and 0.6 ± 0.1 µmol/L in patients with an *IDH1* wild-type tumor, respectively (*p* = 0.0008 and *p* = 0.0006; Figure 3). The *D/L*-2HG ratio was significantly higher in patients with an *IDH1*-mutated tumor compared to patients with an *IDH1* wild-type tumor; 20.6 (95% confidence interval [CI] 8.6–32.5) versus 1.83 (95% CI 1.4–2.2; *p* < 0.0001; Figure 4). As illustrated in Figure 4, the optimal cutoff value of the *D/L*-2HG ratio was 4.5 for the presence of an *IDH1* mutation. This cutoff value predicted the presence of an *IDH1* mutation with a sensitivity of 90% and a specificity of 100%. Two patients with an *IDH1*-mutated tumor had a *D/L*-2HG ratio lower than 4.5.

### 3.5. ctDNA

In order to detect *IDH1/2* mutation status in serum, we investigated serum on ctDNA. Since the most prevalent hotspot mutations in *IDH1* are *IDH1^R132H^* and *IDH1^R132C^*, we used these mutations to design digital droplet PCR. Serum of 3 random patients were investigated to detect ctDNA of these *IDH1* mutations. In all the samples, we successfully detected the *IDH1* mutations (Figure 5).

### 3.6. Tumor Responses

At first radiological evaluation after eight weeks of treatment, all patients had progressive disease and discontinued study treatment (Table 3). Since *D*-2HG serves as a surrogate biomarker of progression, we monitored serial *D*-2HG serum concentrations in order to investigate possible biochemical treatment responses. As illustrated in Figure 6, patients treated with dose level 1 and 2 had an increasing *D*-2HG serum concentration over time (not significant, *p* = 0.1 and *p* = 0.23, respectively). However, patients treated with dose level 3 had a stable *D*-2HG serum concentration (no significant difference between doses, *p* = 0.1) but progressive disease at radiological evaluation.

## 4. Discussion

This is the first clinical trial to describe the toxicity profile, safety and pharmacokinetics of the combination of metformin and chloroquine in patients with *IDH1*-mutated chondrosarcoma, glioma and intrahepatic cholangiocarcinoma. We found that the combination regimen of metformin and chloroquine is well tolerated, but the combination did not induce a clinical response in this patient population. On the other hand, our results confirm the clinical usefulness of *D/L*-2HG ratios in serum as a biomarker for the presence of an *IDH1*-mutated solid tumor and the ddPCR-facilitated detection of an *IDH1* mutation in ctDNA from peripheral blood.

The rationale of using metformin and chloroquine in order to disrupt the metabolism of *IDH1*-mutated solid tumors and to inhibit tumor growth was not supported by our clinical data, since ten out of twelve patients showed tumor progression during study treatment. After we published the first randomized controlled trial studying metformin in pancreatic cancer with a survival endpoint [27], dozens of negative clinical trials with metformin in cancer have been published. So far, only two clinical trials have shown a benefit of metformin on progression-free survival in cancer, and they both concerned non-small cell lung cancer [46,47]. The mechanism and pathophysiological background of the sensitivity of metformin in especially non-small cell lung cancer are currently unknown. High intracellular metformin concentrations are needed in order to induce profound metabolic inhibition and this may be unattainable using oral administration. Since metformin failed to show any metabolic or anti-tumor effect in this trial, phenformin should be considered as an alternative, since phenformin is the lipophilic analog of metformin and may have a better intratumoral bioavailability.

In line with previous studies, this study confirms that 2HG serum concentrations serve as a surrogate biomarker of the presence of an *IDH1* mutation. The prediction of the presence of an *IDH1/2* mutation by 2HG measurement is well established in AML [16]. A study by Borger et al. of 18 patients with *IDH1*-mutated intrahepatic cholangiocarcinoma showed a sensitivity of 83% and a specificity of 90% at a cutoff of 2HG serum levels ≥1.15 mmol/L [48]. Our results confirmed this observation and since our method distinguishes the *IDH1* mutation-specific *D*-2HG from the less specific *L*-2HG, we suggest that the *D/L*-2HG ratio performs even better in a more heterogeneous patient population, with a sensitivity of 90% and a specificity of 100% at cutoff of 4.5 for the presence of an *IDH1* mutation. This is in accordance with a report by Delahousse et al. including 8 patients with *IDH1*-mutated intrahepatic cholangiocarcinoma and 9 patients with wild-type *IDH1* intrahepatic cholangiocarcinoma, which proposed a *D/L*-2HG ratio cutoff of 4.9 [49]. With three independent studies supporting the use of *D*-2HG measurement as a pre-screening tool for *IDH1* mutational status with high accuracy and precision, this technique fulfils the criteria for implementation in routine clinical use.

Since the presence of *IDH1* mutations is relevant for prognosis and treatment, 2HG may serve as a surrogate marker of treatment efficacy. Levels of 2HG in serum and urine of patients with *IDH1/2*-mutated AML decreased throughout conventional therapy, concordant with a decrease in blast counts. However, the concentration of serum 2HG is substantially lower in patients with an *IDH1*-mutated solid tumors compared with patients with *IDH1/2*-mutated AML (5.6 µmol/L in our study versus 21.2 µmol/L in AML) [16]. We observed increasing *D*-2HG serum concentrations in low dose-level treatment, but a stable concentration of *D*-2HG in the highest dose-level of metformin in combination with chloroquine. Future studies are needed to determine how well *D*-2HG correlates with changes in tumor volume during treatment.

We were able to confirm the clinical utility of ctDNA in *IDH1*-mutated cancers as described earlier [50]. The use of plasma-derived ctDNA is a promising tool for treatment decision-making based on predictive testing, detection of resistance mechanisms, and monitoring tumor response. By utilization of digital droplet PCR, we were able to detect tumor-specific *IDH1* hotspot mutations in ctDNA, which may facilitate the monitoring of tumor response during therapy. In addition, our data generate the hypothesis that the longitudinal evaluation of *D/L*-2HG ratios can be used to determine effects of anti-cancer treatment, although the stable *D/L*-2HG ratios in the highest dose level did not corroborate with clinical responses and the number of studied patients was small. Necessary steps for the translation of these minimally invasive measurements to clinical practice is subject for future research.

## 5. Conclusions

Results from this prospective, open-label, phase Ib study show that the combination of metformin and chloroquine has a favorable toxicity profile but no clinical activity in patients with *IDH1*-mutated chondrosarcoma, glioma and intrahepatic cholangiocarcinoma. In addition, our data confirm and support the use of *D*-2HG measurements as a screening tool for *IDH1* mutational status in routine clinical use for these tumors. Although our analyses of tumor responses and overall survival are based on very small numbers and late-stage cancer patients, alternative combination regimens disrupting the metabolism in *IDH1/2*-mutated cancers should be investigated in future studies.

## Figures and Tables

**Figure 1 cancers-13-02474-f001:**
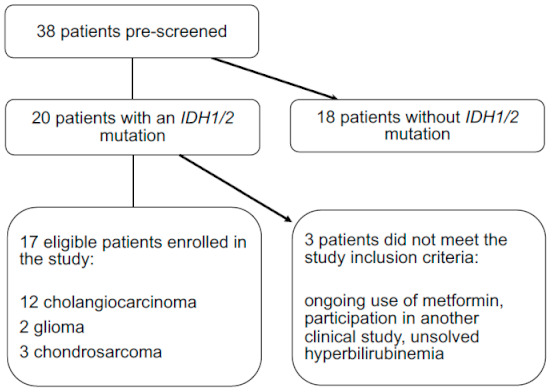
Flowchart showing that 38 patients were pre-screened for the presence of an *IDH1/2* mutation. Twenty patients had an *IDH1* mutation and 17 patients met the study inclusion criteria.

**Figure 2 cancers-13-02474-f002:**
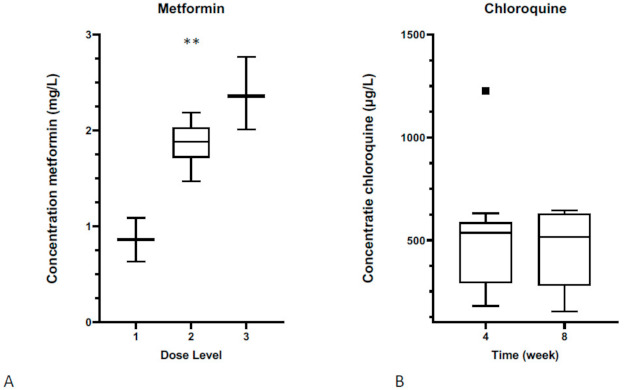
Serum concentrations of metformin and chloroquine. (**A**) A dose level-dependent increase of the plasma metformin concentration is shown; (** *p* < 0.01, one-way ANOVA test). (**B**) Non-escalating chloroquine doses in time, resulting in stable plasma concentrations at subsequent time points.

**Figure 3 cancers-13-02474-f003:**
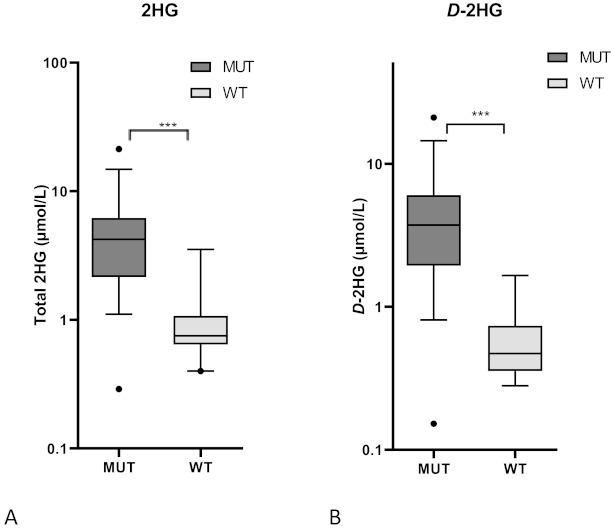
Serum concentrations of (**A**) total 2HG and (**B**) *D*-2HG was 5.6 ± 1.3 µmol/L and 5.3 ± 1.2 µmol/L in patients with an *IDH1*-mutated tumor and 1.0 ± 0.2 µmol/L and 0.6 ± 0.1 µmol/L in patients with an *IDH1* wild-type tumor, respectively. (*** *p* < 0.001, two-way Mann–Whitney test).

**Figure 4 cancers-13-02474-f004:**
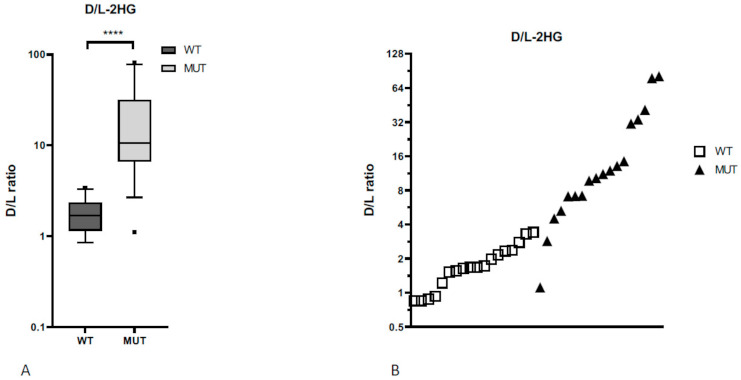
The *D/L*-2HG ratio (**A**) was significantly higher in patients with an *IDH1* mutation compared to patients without an *IDH1* mutation (**** *p* < 0.0001, two-way Mann–Whitney test). (**B**) The optimal cutoff value of 4.5 for the presence of an *IDH1* mutation is shown.

**Figure 5 cancers-13-02474-f005:**
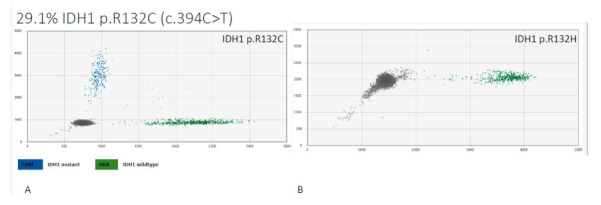
Serum analysis of ctDNA by NGS to detect hotspots mutations *IDH1*^R132C^ and *IDH1*^R132H^. Analysis of one patient sample (**A**) with blue spots indicating detection of *IDH1*^R132C^ ctDNA that consists of 29% ctDNA, (**B**) *IDH1*^R132H^ was not detected.

**Figure 6 cancers-13-02474-f006:**
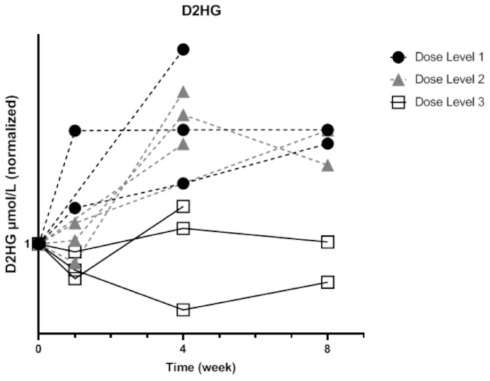
Serial *D*-2HG serum concentrations of patients treated with dose levels 1, 2 and 3. *D*-2HG serum concentrations were increased with time in dose levels 1 and 2 (not significant, respectively *p* = 0.1 and *p* = 0.23, Kruskal–Wallis test), whereas with dose level 3, stable or lower *D*-2HG serum concentrations were found (Not significant, *p* = 0.1, one-way ANOVA test).

**Table 1 cancers-13-02474-t001:** Dose-escalation schedule for metformin and chloroquine.

Dose Level	Dose of Metformin (Total Daily Dose)	Dose of Chloroquine (Total Daily Dose)
−1	500 mg once a day (500 mg total)	200 mg once a day
1	500 mg two times a day (1000 mg total)	200 mg once a day
2	1000 mg two times a day (2000 mg total)	200 mg once a day
3	1500 mg two times a day (3000 mg total)	200 mg once a day

**Table 2 cancers-13-02474-t002:** Patient demographics and disease characteristics. Chemotherapy regimens were given in an advanced setting, unless stated otherwise.

Pt #	Gender	Age	WHO-PS	Primary Diagnosis	Earlier Surgery	Earlier Systemic Therapy	Earlier Radiotherapy	Time Since Initial Diagnosis
								(in Years)
1	Male	54	0	Chondrosarcoma	Resection of tumor of the right knee	None	No	5.6
2	Male	49	1	Cholangiocarcinoma	Right hemihepatectomy	Gemcitabine/cisplatin	No (during study)	3.92
3	Male	51	2	Chondrosarcoma	Resection of left pelvic tumor	Sirolimus/cyclophosphamide	Yes	3.25
4	Male	64	0	Cholangiocarcinoma	Exploratory laparotomy	Gemcitabine/cisplatin	No	3.3
5	Male	58	1	Cholangiocarcinoma	Right hemihepatectomy	Gemcitabine/cisplatin	No	3.72
6	Male	57	1	Cholangiocarcinoma	None	Gemcitabine/cisplatin	No	2.84
7	Male	82	2	Cholangiocarcinoma	None	Gemcitabine/cisplatin, pembrolizumab	Yes	3.75
8	Female	53	1	Cholangiocarcinoma	None	Gemcitabine/cisplatin	No	2.33
9	Male	70	0	Cholangiocarcinoma	Right hemihepatectomy	Gemcitabine/cisplatin, capecitabine/oxaliplatin	No	2.58
10	Male	39	0	Cholangiocarcinoma	Right hemihepatectomy	Gemcitabine/cisplatin, folfirinox	No	3.82
11	Male	50	0	Cholangiocarcinoma	None	Gemcitabine/cisplatin, gemcitabine/oxaliplatin	No	3.25
12	Male	39	0	Chondrosarcoma	Resection of right scapular tumor	Sirolimus/cyclophosphamide	No	2.96
13	Female	42	0	Cholangiocarcinoma	None	Gemcitabine/cisplatin	Yes	1.53
14	Male	46	1	Glioma	Tumor resection	Temozolomide	Yes	6.43
15	Male	34	1	Glioma	Tumor resection right frontal	Temozolomide, lomustine	Yes	5.33
16	Female	63	0	Cholangiocarcinoma	Right hemihepatectomy	Gemcitabine/cisplatin	No	2.50
17	Male	64	0	Cholangiocarcinoma	None	Gemcitabine/cisplatin, CAPOX	No	1.50

Abbreviations: Pt #, patient number; WHO-PS, World Health Organization- Performance Status; CAPOX, Capecitabine/Oxaliplatin.

**Table 3 cancers-13-02474-t003:** Description of administered doses, dose-limiting toxicities and serious adverse events.

Pt #	Metformin Dose (mg)	Chloroquine Dose (mg)	DLT (Grade)	SAE (Grade)	Days on Study	Reason for Study Termination	Overall Survival
							(Days after Start of Study)
1	1000	200	-	-	61	Progressive disease (CT)	818
2	1000	200	-	-	33	Patient decision (toxicity)	66
3	1000	-	-	-	7	Progressive disease (clinical)	29
4	1000	200	-	-	56	Progressive disease	426
5	2000	200	-	-	33	Progressive disease (clinical and CT)	951
6	2000	200	-	-	43	Patient decision (toxicity)	351
7	1000	200	-	-	14	Progressive disease (clinical)	680
8	2000	200	-	-	17	Patient decision (toxicity)	322
9	2000	200	-	-	62	Progressive disease (CT)	194
10	2000	200	-	-	58	Progressive disease (CT)	108
11	2000	200	-	-	67	Progressive disease (CT)	323
12	3000	200	-	-	59	Progressive disease (CT)	330
13	2000	200	-	-	59	Progressive disease (CT)	255
14	3000	200	-	Hydrocephalus (4)	28	Progressive disease (clinical)	154
15	3000	-	-	-	13	Progressive disease (clinical)	42
16	3000	200	-	Bile duct stenosis (3)	43	Progressive disease (CT)	92
17	3000	200	-	-	74	Progressive disease (CT)	102

**Table 4 cancers-13-02474-t004:** Possible, probable or definitive treatment-related adverse event.

	Dose Level 1		Dose Level 2		Dose Level 3	
Number of Patients:	*n* = 3		*n* = 6		*n* = 3	
CTCAE Grade:	1–2	3–4	1–2	3–4	1–2	3–4
Fatigue	1		3		2	
Anorexia	1		6		2	
Nausea	3		6		2	
Vomiting			2		2	
Diarrhea			4		1	
Constipation	1		1			
Weight loss					1	
Abdominal pain			1		1	

Numbers represent number of patients. Abbreviation: CTCAE, Common Terminology Criteria for Adverse Events version 4.0.

## Data Availability

The data presented in this study are available on request from the corresponding author.

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
