# Peer review of "A Phase Ib Clinical Trial of Metformin and Chloroquine in Patients with IDH1-Mutated Solid Tumors"

_cancers, 2021, doi:10.3390/cancers13102474_

Round 1
Reviewer 1 Report
A well written manuscript with useful information in the form of clinical trial results for treatment of 17 IDH1 mutant glioma/cholangiocarcinoma/ chondrosarcoma patients with oral antidiabetic metformin and the antimalarial drug chloroquine. The aims were to identify whether the drugs disrupted solid tumour metabolism leading to therapeutic effects /dose efficacy and support evidence of clinical usefulness of D/L 2-HG ratios as a biomarker screening tool.
The introduction could be improved:
In the intro it might be worth including some mention of clinically approved IDH inhibitors and/or those currently in clinical trials for both AML and glioma/others?
The discussion of glutaminolysis – e.g. paragraph starting line 78 discussing IDH1/2 mutated cells being dependent on glutaminolysis for TCA cycle anaplerosis via a-KG for 2-HG production. Conflating production of 2-HG in IDH1 and IDH2 mutant cells as written is perhaps not justified given the mechanisms are different and the process takes part in different compartments of the cell – e.g. mitochondria vs cytoplasm. The references given are only partially relevant to the importance of glutaminolysis and do support a common mechanism as implied here. Much less is known about IDH2 metabolism yet this is the mechanism implicated in the description here e.g. 2-HG production via TCA cycle-derived 2-HG. Suggest this section, implicated as it is in justifying why chloroquine was used in the study, be more clearly and accurately presented?
No mention of in-vitro models/animal model for metformin treatment or relevant IDH1 mutant studies E.g. possible some refs missing for example:
https://www.mdpi.com/2072-6694/12/1/210/htm
https://www.mdpi.com/2072-6694/11/12/1895
The results sections could be improved:
As stated in line 214-15 no patients with IDH2 mutations took place in the trial. However, in a number of places the results refer to being relevant to both ‘IDH1/2’. For example, in the title of the paper, Figure 1, Line 272, caption Fig4 – check throughout manuscript and remove redundant references to IDH2 (much less is known about IDH2 compared to 1 and the metabolic effects and mechanisms of this IDH2 mutations should not be conflated with those of IDH1 by implication).
Section 3.4 – suggest use term ‘more specific’ or less specific’ to refer to D-2HG / L-2HG in relation to IDH mutant cells as your data (and others) show D-2HG is not entirely specific to IDH mutations in the way discussed.
Line 273 - details of the 38 patients screened are relevant given you are using their data to make predictions about 2-HG as a potential diagnostic marker. Suggest including relevant clinical/non-clinal (patient demographics/disease characteristics as table in SI or extension of table 2 in SI.
The methods sections could be improved:
Generally, well described but lacks some accuracy. e.g. the description of the enantiomeric 2-HG measurement is misleading: Line 175-176 states the enantiomeric forms of 2-HG were detected by MS which unlikely given the enantiomers are isobaric and hence indistinguishable by MS alone. Its likely the measurements were conducted using a hyphenated technique such as LC-MS or GC-MS and with the important functional component being the chromatography. Accuracy important given the attention given to D-2HG as a diagnostic/prognostic indicator in the discussion section.
The conclusions are generally supported by the results but some improvement could be made:
In particular as stated in line 214-15 no patients with IDH2 mutations took place in the trial. However, in a number of places the conclusions refer to being relevant to both ‘IDH1/2’ – such conclusions are not supported by the results and should be avoided. E.g. line 350, 374 etc (need to be checked throughout).
Author Response
REVIEWER 1 COMMENTS
Comment 1: The introduction could be improved:
(I) In the intro it might be worth including some mention of clinically approved IDH inhibitors and/or those currently in clinical trials for both AML and glioma/others?
(II)The discussion of glutaminolysis – e.g. paragraph starting line 78 discussing IDH1/2 mutated cells being dependent on glutaminolysis for TCA cycle anaplerosis via a-KG for 2-HG production. Conflating production of 2-HG in IDH1 and IDH2 mutant cells as written is perhaps not justified given the mechanisms are different and the process takes part in different compartments of the cell – e.g. mitochondria vs cytoplasm. The references given are only partially relevant to the importance of glutaminolysis and do support a common mechanism as implied here. Much less is known about IDH2 metabolism yet this is the mechanism implicated in the description here e.g. 2-HG production via TCA cycle-derived 2-HG.
(III)Suggest this section, implicated as it is in justifying why chloroquine was used in the study, be more clearly and accurately presented?
(IV) No mention of in-vitro models/animal model for metformin treatment or relevant IDH1 mutant studies E.g. possible some refs missing for example: https://www.mdpi.com/2072-6694/12/1/210/htm https://www.mdpi.com/2072-6694/11/12/1895
Response to Reviewer
We appreciate the comments of the reviewer in response to the introduction section; with respect to
(I) The clinical trials of IDH1/2-mutant inhibitors: we added a section to the introduction in which we describe clinically approved IDH1/2-mutant inhibitors.
(II) According to the mentioned paragraph discussing the glutaminolysis in IDH1/2 mutated cells, we agree with the reviewer that the statement of IDH1/2 mutated cells depending on glutaminolysis for TCA cycle anaplerosis is only justified for IDH1-mutated cells. Since the metabolism reprogramming caused by IDH2 mutations is less investigated, we refer only to the better explored metabolism of IDH1 mutant cells.
(III) The importance of chloroquine in the treatment is first presented (line 86) as potential inhibitor of the final step of glutaminolysis targeting glutamate dehydrogenase. The second property of chloroquine by inhibiting autophagy is described in line 90. As noted, the combination may thus be selective for IDH1/2-mutated cells because it inhibits glutaminolysis and autophagy on which these cells depend.
(IV) We added an additional paragraph describing models wherein IDH1-mutated cancer cells are more sensitive to metformin compared to IDH1 wild-type cancer cells. In addition, we added the missing and mentioned references by the reviewer.
Comment 2: The results sections could be improved:
(I) As stated in line 214-15 no patients with IDH2 mutations took place in the trial. However, in a number of places the results refer to being relevant to both ‘IDH1/2’. For example, in the title of the paper, Figure 1, Line 272, caption Fig4 – check throughout manuscript and remove redundant references to IDH2 (much less is known about IDH2 compared to 1 and the metabolic effects and mechanisms of this IDH2 mutations should not be conflated with those of IDH1 by implication).
(II) Section 3.4 – suggest use term ‘more specific’ or less specific’ to refer to D-2HG / L-2HG in relation to IDH mutant cells as your data (and others) show D-2HG is not entirely specific to IDH mutations in the way discussed.
(III) Line 273 - details of the 38 patients screened are relevant given you are using their data to make predictions about 2-HG as a potential diagnostic marker. Suggest including relevant clinical/non-clinal (patient demographics/disease characteristics as table in SI or extension of table 2 in SI.
Response to Reviewer
We appreciate the comments of the reviewer in response to the results section;
(I) With respect to remarks that no patients with an IDH2-mutated cancer were included in the trial, we agree with the reviewer that the results should be more focused on findings according included IDH1-mutated patients. Moreover, we adjusted and removed references to IDH2 mutations where necessary.
(II) We agree with the reviewer to use “more and less specific” to refer to D-2HG/L-2HG in relation to IDH mutant cells and changed this in the manuscript.
(III) From all 38 screened patients the D/L-2HG ratio was analyzed and NGS was performed for IDH1/2 mutation status. As suggested by the reviewer, we included the most relevant clinical information (the primary diagnoses) of the patients with IDH1 wild-type and added this to the results sections.
Comment 3: The methods sections could be improved:
Generally, well described but lacks some accuracy. e.g. the description of the enantiomeric 2-HG measurement is misleading: Line 175-176 states the enantiomeric forms of 2-HG were detected by MS which unlikely given the enantiomers are isobaric and hence indistinguishable by MS alone. Its likely the measurements were conducted using a hyphenated technique such as LC-MS or GC-MS and with the important functional component being the chromatography. Accuracy important given the attention given to D-2HG as a diagnostic/prognostic indicator in the discussion section.
Response to Reviewer
We appreciate the comments of the reviewer in response to the methods section and totally agree that the annotation according to the technique we used for detecting of D-2HG and L-2HG is missing. As added to the manuscript and indicated by the reviewer, we used liquid chromatography-tandem mass spectrometry (LC-MS/MS). Our laboratory is capable of enantiomer-specific detection of D-2HG and L-2HG using (+)-Di-O-acetyl-l-tartaric anhydride (DATAN), a chiral derivatizing agent, to modify the D and L-stereoisomers of 2HG, allowing separation and quantification by LC-MS/MS, as we described earlier in https://pubmed.ncbi.nlm.nih.gov/28735490/ and https://pubmed.ncbi.nlm.nih.gov/15166110/
Comment 4: The conclusions are generally supported by the results but some improvement could be made:
In particular as stated in line 214-15 no patients with IDH2 mutations took place in the trial. However, in a number of places the conclusions refer to being relevant to both ‘IDH1/2’ – such conclusions are not supported by the results and should be avoided. E.g. line 350, 374 etc (need to be checked throughout).
Response to Reviewer
We appreciate the comments of the reviewer and as discussed earlier, we agree with the reviewer that the results and conclusions should be more focused on IDH1-mutated patients. We adjusted and removed redundant references to IDH2 mutations where necessary.

Reviewer 2 Report
Due to lack of effective treatment options in high-grade chondrosarcomas, gliomas and intrahepatic cholangiocarcinomas, these aggressive types of cancer still remain devastating diseases worldwide.
In the manuscript entitled “A phase Ib clinical trial of metformin and chloroquine in patients with IDH1-mutated or IDH2-mutated solid tumors”, the Authors present the results from phase Ib clinical trial investigating the safety and maximum tolerated dose (MTD) of combined treatment with metformin and chloroquine in patients with IDH1-mutated chondrosarcoma, glioma and intrahepatic cholangiocarcinoma. What is more, the Authors confirmed the usefulness of D/L-2HG ratios in serum as a biomarker for the presence of an IDH1-mutated solid tumors and the digital droplet PCR-facilitated detection of an IDH1 mutation in circulating DNA (ctDNA) from peripheral blood. Therefore, they provided the rationales for the implementation in routine clinical use.
Nevertheless, in my opinion, some issues need to be addressed and explained.
Major concerns:
- I do not fully agree with the Authors’ statement: “The prognosis of these tumors remains poor, irrespective of the IDH1/2 mutation status” (line 48). Many studies showed that patients with IDH-mutated gliomas have better survival compared to their IDH-wild type counterparts irrespective of histology and grade, making mutations in IDH the most essential prognostic factor for survival, followed by age, tumor grade, and MGMT gene status [Madala, H.R.; Punganuru, S.R.; Arutla, V.; Misra, S.; Thomas, T.J.; Srivenugopal, K.S. Beyond Brooding on Oncometabolic Havoc in IDH-Mutant Gliomas and AML: Current and Future Therapeutic Strategies. Cancers, 2018, 10, 49]. Clinical statistics show that a median overall survival (OS) is 31 months for secondary GBM patients with IDH mutations compared to 15 months for those without the mutations. Patients with IDH-mut anaplastic astrocytoma have 65 months of median OS compared to 20 months in IDH-wt [Yan, H.; Parsons, D.W.; Jin, G.; McLendon, R.; Rasheed, B.A.; Yuan, W.; Kos, I.; Batinic-Haberle, I.; Jones, S.; Riggins, G.J.; et al. IDH1 and IDH2 mutations in gliomas. Engl. J. Med. 2009, 360, 765–773].
- The main weak point of this study is that no patients with an IDH2 mutation were enrolled. However, the mutations in IDH2 are more common in haematological tumors and relatively rare in solid tumors (not as frequent as IDH1), so there could have been some difficulties in completing cohorts with mutant IDH2. Following this issue, since there are no patients with mutations in IDH2 enrolled, I would recommend the Authors not to use the “IDH1/2 mutations/ IDH1/2-mutated” phrases within the text (apart from the Introduction) but just only “IDH1 mutations/ IDH1-mutated” etc.
- Supplementary Figure S1 – were any of the differences significant or is there no significance between single-agent administration (week 4) and co-administration with chloroquine (week 8)? There is no information in the figure legend how the statistic was calculated (type of statistical test).
- Figure 3 and Figure 4 – there is no information in the figures legend how the statistic was calculated (what type of statistical test?).
- It is unclear why only 3 patients were investigated to detect ctDNA and mutations in IDH1 by ddPCR. Maybe using the serum from all eligible patients enrolled in the study would be more informative?
- Figure 6 - D-2HG serum concentrations are increased with time in dose levels 1 & 2, whereas with dose level 3, stable or lower. Were any of the differences (within the dose, and between the doses) between week 4 and 8 statistically significant? There is no information in the figure legend how the statistic was calculated (what type of statistical test?)
- How could the Authors explain the findings that patients treated with low doses level (1&2) followed by increased D-2HG serum conc. have progressive disease and patients treated with the highest dose (level 3) had a stable/lower D-2HG serum conc. but also clinically progressed? Would the Authors expect that there should be a correlation between the level of D-2HG in serum and the patients outcome (progression or stable disease)?
- Seeing that the treatment of advanced IDH1-mutated solid tumors with metformin and chloroquine was well tolerated however, no metabolic or anti-tumor effects were found (all patients showed clinical/ radiological progression after 8 weeks of treatment), is there (in opinion of the Authors) the rationale for further follow-up phase II clinical study? If possible, there was no clinical responses in this patient cohort since the number of patients enrolled was too small (e.g. only 2 patients with glioma and 3 with chondrosarcoma) and not sufficient to clearly observe and distinguish the real metabolic or anti-cancer effects of the treatment of these types of tumor.
- Interestingly, the Authors mention phenformin as the lipophilic analog of metformin and its possible better intratumoral bioavailability. Are there any reports in the literature showing significantly better (intratumoral) bioavailability of phenformin in any (cancer) in vivo animal model?
Minor concerns:
- Within the Materials & methods section the Authors state that therapy response was assessed by imaging with computed tomography (CT) or magnetic resonance imaging (MRI) scans. However, in the Table 3 there is information only about the CT imaging used for confirmation of progressive disease, but no MRI. And, what about the patient #4 – there is no information in the Table 3?
- Within the text there are mixed different types of fonts and some typos are found.
- Lines 174-177: the text should be reorganised because of some repeats and reduplications.
Author Response
REVIEWER 2 COMMENTS
Major concerns:
- I do not fully agree with the Authors’ statement: “The prognosis of these tumors remains poor, irrespective of the IDH1/2 mutation status”(line 48). Many studies showed that patients with IDH-mutated gliomas have better survival compared to their IDH-wild type counterparts irrespective of histology and grade, making mutations in IDH the most essential prognostic factor for survival, followed by age, tumor grade, and MGMT gene status [Madala, H.R.; Punganuru, S.R.; Arutla, V.; Misra, S.; Thomas, T.J.; Srivenugopal, K.S. Beyond Brooding on Oncometabolic Havoc in IDH-Mutant Gliomas and AML: Current and Future Therapeutic Strategies. Cancers, 2018, 10, 49]. Clinical statistics show that a median overall survival (OS) is 31 months for secondary GBM patients with IDH mutations compared to 15 months for those without the mutations. Patients with IDH-mut anaplastic astrocytoma have 65 months of median OS compared to 20 months in IDH-wt [Yan, H.; Parsons, D.W.; Jin, G.; McLendon, R.; Rasheed, B.A.; Yuan, W.; Kos, I.; Batinic-Haberle, I.; Jones, S.; Riggins, G.J.; et al. IDH1 and IDH2 mutations in gliomas. J. Med. 2009, 360, 765–773].
Response to Reviewer
We appreciate the comments of the reviewer with respect to the prognosis of patients with IDH mutations. We agree that glioma patients with an IDH1 mutation have a relatively prolonged survival compared to IDH-wild counterparts and adjusted the statement in the manuscript.
- The main weak point of this study is that no patients with an IDH2 mutation were enrolled. However, the mutations in IDH2are more common in haematological tumors and relatively rare in solid tumors (not as frequent as IDH1), so there could have been some difficulties in completing cohorts with mutant Following this issue, since there are no patients with mutations in IDH2 enrolled, I would recommend the Authors not to use the “IDH1/2 mutations/ IDH1/2-mutated” phrases within the text (apart from the Introduction) but just only “IDH1 mutations/ IDH1-mutated” etc.
Response to Reviewer
We appreciate the comments with respect to remarks that no patients with an IDH2 mutation were included in the trial. We agree that the results and conclusions should be more focused on IDH1-mutated patients. We have adjusted this throughout the manuscript and removed redundant references to IDH2 mutations where necessary.
- Supplementary Figure S1 – were any of the differences significant or is there no significance between single-agent administration (week 4) and co-administration with chloroquine (week 8)? There is no information in the figure legend how the statistic was calculated (type of statistical test).
Response to Reviewer
There was indeed no significant difference between single-agent administration (week 1) and co-administration with chloroquine (week 4 and 8). Additional statistical information is added to the legend.
- Figure 3 and Figure 4 – there is no information in the figures legend how the statistic was calculated (what type of statistical test?).
Response to Reviewer
We appreciate the comment notifying the missing information, we added statistical information to the figures’ legend.
- It is unclear why only 3 patients were investigated to detect ctDNA and mutations in IDH1by ddPCR. Maybe using the serum from all eligible patients enrolled in the study would be more informative?
Response to Reviewer
We agree that using serum to detect ctDNA from all eligible patients would be much more informative, but the serum of the patients was first used for assays needed for the primary and secondary objectives. Unfortunately, there were only 3 patients of whom we had a sufficient amount of serum left to use for detection of ctDNA.
- Figure 6 - D-2HG serum concentrations are increased with time in dose levels 1 & 2, whereas with dose level 3, stable or lower. Were any of the differences (within the dose, and between the doses) between week 4 and 8 statistically significant? There is no information in the figure legend how the statistic was calculated (what type of statistical test?)
Response to Reviewer
There was no significant difference within dose level 1 (P=0.1), level 2 (P=0.23) and between the doses in week 1-4-8 (P=0.1). Additional statistical information is added to the legend of the figure and the manusscript.
- How could the Authors explain the findings that patients treated with low doses level (1&2) followed by increased D-2HG serum conc. have progressive disease and patients treated with the highest dose (level 3) had a stable/lower D-2HG serum conc. but also clinically progressed? Would the Authors expect that there should be a correlation between the level of D-2HG in serum and the patients outcome (progression or stable disease)?
Response to Reviewer
The hypothesis of this trial was that by the combination of anti-metabolic drugs, the metabolism of IDH1-mutated cancers cells and their production of D-2HG would be inhibited. Although the findings are not significant, the pattern we describe indicates that D-2HG may serve as a surrogate marker of treatment efficacy. By first confirming the technique and implementation of D-2HG measurements in the clinic, future studies have to determine how well D-2HG correlates with changes in tumor volume during the course of treatment.
- Seeing that the treatment of advanced IDH1-mutated solid tumors with metformin and chloroquine was well tolerated however, no metabolic or anti-tumor effects were found (all patients showed clinical/ radiological progression after 8 weeks of treatment), is there (in opinion of the Authors) the rationale for further follow-up phase II clinical study? If possible, there was no clinical responses in this patient cohort since the number of patients enrolled was too small (e.g. only 2 patients with glioma and 3 with chondrosarcoma) and not sufficient to clearly observe and distinguish the real metabolic or anti-cancer effects of the treatment of these types of tumor.
Response to Reviewer
The rationale of using metformin and chloroquine in order to disrupt the metabolism of IDH1/2-mutated solid tumors and inhibit progression, could not be supported by our clinical data. This may be the result of an incorrect hypothesis that solid IDH1-mutated tumors are susceptible to anti-metabolic drugs, or due to a type 2 error; either due to our small sample size or because the studied cancers were too late-stage to be affected by the anti-cancer effects of metformin, if any. An alternative explanation may be offered by the poor bio-availability of the hydrophilic metformin, which may be circumvented by using its lipophilic analogue phenformin (see below).
- Interestingly, the Authors mention phenformin as the lipophilic analog of metformin and its possible better intratumoral bioavailability. Are there any reports in the literature showing significantly better (intratumoral) bioavailability of phenformin in any (cancer) in vivoanimal model?
Response to Reviewer
In vitro, high metformin concentrations (mM) are needed for profound ETC inhibition and may be unattainable. Due to its lipophilicity, phenformin may be more effective than metformin. Phenformin already has shown anti-cancer efficacy against IDH mutated cells and many other cancer (stem) cells in vitro at low μM concentrations. https://pubmed.ncbi.nlm.nih.gov/32049007, https://pubmed.ncbi.nlm.nih.gov/23352126/
Minor concerns:
- Within the Materials & methodssection the Authors state that therapy response was assessed by imaging with computed tomography (CT) or magnetic resonance imaging (MRI) scans. However, in the Table 3 there is information only about the CT imaging used for confirmation of progressive disease, but no MRI. And, what about the patient #4 – there is no information in the Table 3?
Response to Reviewer
In order to determine therapy response/progressive disease; CT was used for patients with cholangiocarcinoma and chrondrosarcoma and MRI was used for patients with glioma. However, as described in table 3 both glioma patients had clinical disease progression before assessing MRI in week 8. According information of patient #4 in table 3, no missing data is noticed.
- Within the text there are mixed different types of fonts and some typos are found.
Response to Reviewer: the manuscript is corrected for different types of fonts
- Lines 174-177: the text should be reorganised because of some repeats and reduplications.
Response to Reviewer: unnecessary repeats are removed and text is reorganized.

Round 2
Reviewer 2 Report
In the revised version of the manuscript the Authors have taken the comments and suggestions into consideration and have significantly improved their manuscript. The Author have addressed and clarified all major and minor concerns and strengthened the manuscript. In my opinion, this manuscript can be published in Cancers.